# Extracellular vesicles from human plasma and serum are carriers of extravesicular cargo— Implications for biomarker discovery

Mari Palviainen[1,2,3], Mayank Saraswat[4,5], Zoltán Varga[6], Diána Kitka[6], Maarit Neuvonen[1,2,3], Maija Puhka[2,7], Sakari Joenväärä[4,5], Risto Renkonen[4,5], Rienk Nieuwland[8,9], Maarit Takatalo[1,2,3], Pia R. M. Siljander[1,2,3] *

1 EV group, Molecular and Integrative Biosciences Research Programme, Faculty of Biological and Environmental Sciences, University of Helsinki, Helsinki, Finland, 2 EV-core, University of Helsinki, Helsinki, Finland, 3 CURED, Drug Research Program, Faculty of Pharmacy, Division of Pharmaceutical Biosciences, University of Helsinki, Helsinki, Finland, 4 Transplantation Laboratory, Haartman Institute, University of Helsinki, Helsinki, Finland, 5 HUS Diagnostic Center, Helsinki University Hospital, Helsinki, Finland, 6 Biological Nanochemistry Research Group, Research Centre for Natural Sciences, Institute of Materials and Environmental Chemistry, Budapest, Hungary, 7 Institute for Molecular Medicine Finland FIMM, University of Helsinki, Helsinki, Finland, 8 Laboratory Experimental Clinical Chemistry, Amsterdam UMC, University of Amsterdam, Amsterdam, The Netherlands, 9 Vesicle Observation Center, Amsterdam UMC, University of Amsterdam, Amsterdam, The Netherlands

* pia.siljander@helsinki.fi

## Abstract

Extracellular vesicles (EVs) in human blood are a potential source of biomarkers. To which extent anticoagulation affects their concentration, cellular origin and protein composition is largely unexplored. To study this, blood from 23 healthy subjects was collected in acid citrate dextrose (ACD), citrate or EDTA, or without anticoagulation to obtain serum. EVs were isolated by ultracentrifugation or by size-exclusion chromatography (SEC) for fluorescence-SEC. EVs were analyzed by micro flow cytometry, NTA, TEM, Western blot, and protein mass spectrometry. The plasma EV concentration was unaffected by anticoagulants, but serum contained more platelet EVs. The protein composition of plasma EVs differed between anticoagulants, and between plasma and serum. Comparison to other studies further revealed that the shared EV protein composition resembles the "protein corona" of synthetic nanoparticles incubated in plasma or serum. In conclusion, we have validated a higher concentration of platelet EVs in serum than plasma by contemporary EV methods. Anticoagulation should be carefully described (i) to enable study comparison, (ii) to utilize available sample cohorts, and (iii) when preparing/selecting biobank samples. Further, the similarity of the EV protein corona and that of nanoparticles implicates that EVs carry both intravesicular and extravesicular cargo, which will expand their applicability for biomarker discovery.

## Introduction

Biomarkers are sought as measurable indicators of pathological processes and for treatment monitoring. Extracellular vesicles (EVs) contain a rich and multifaceted cargo of proteins,

**Data Availability Statement:** All proteomic data files are available from the PRIDE database (accession number PXD015375).

**Funding:** This study has been supported by the Academy of Finland (grants 287089 (PRMS) and 315227(M. Palviainen)), Magnus Ehrnrooth Foundation (PRMS), Medicinska Understödsföreningen Liv och Hälsa rf. (PRMS) and by the National Research, Development and Innovation Office NKFIH, Hungary (ZV, grants PD 121326 and NVKP_16-1-2016-0007). ZV was supported by the János Bolyai Research Fellowship. DK was supported by the ÚNKP-19-3 New National Excellence Program of the Ministry for Innovation and Technology.

**Competing interests:** The authors have declared that no competing interests exist.

metabolites, lipids and nucleic acids for biomarker discovery, and therefore EVs are an excellent source of potential biomarkers. Although EVs can be isolated from all body fluids and from tissues, blood provides the richest source of EVs. Blood-borne EVs are a heterogeneous population originating from blood cells, endothelial cells, and for example in cancer patients from tumor cells [1]. Therefore, blood EVs may well serve as biomarkers of multiple diseases, either by their concentration or by a selection of molecules derived from their biochemical composition. So far, the majority of EV biomarker studies have concentrated on the discovery of cancer biomarkers [2], closely followed by biomarkers for cardiovascular diseases [3]. Plasma and serum samples offer in the form of liquid biopsy a relatively non-invasive means to detect disease-related biomarkers making them an optimal target for studies using large study cohorts and biobanks. Unfortunately, most of the currently available biobank samples have not been collected with EVs in mind. Although some confounders of sample uniformity such as anticoagulation, agitation and delay in processing (causing *ex vivo* platelet activation followed by vesiculation) are becoming better acknowledged, currently many preanalytical steps remain unaddressed in a systematic way [4, 5]. Recently, the International Society of Extracellular Vesicles (ISEV) standardization taskforce has realized the urgency of standardizing blood sample collection and generated harmonization initiatives for the field [6].

So far, only a few studies have addressed the effect of anticoagulation–or its absence i.e. serum —on the resulting EV population. The seminal study of George et al. demonstrated that the concentration of "platelet microparticles", isolated by centrifugation at 35 000 x g, were higher in serum than in plasma anticoagulated by acid citrate dextrose (ACD), which was attributed to platelet activation during clot formation [7]. In the study of György et al., ACD was shown to be superior in comparison to citrate in preventing artefactual platelet activation and vesiculation during blood collection [5]. However, there are no studies yet on the effect of anticoagulants on the molecular composition of blood-borne EVs. Therefore, we studied EVs isolated from blood anticoagulated with the three most commonly used anticoagulants and serum by contemporary EV analytics and following the latest recommendations regarding blood handling [6, 8].

In pharmaceutics, nanoparticles (NPs) are developed for drug delivery solutions (DDS). Due to their fast elimination from the body, synthetic NP and their protein corona have recently been under intensive investigation [9–11]. In addition to DDS applications, NP corona has also been suggested to be a future diagnostic tool for e.g. cancer diagnostics [12–14]. Previous studies have shown that the protein corona of NPs can be divided into a hard and a soft part, depending on the binding affinities of the associating proteins [15–17]. Proteins that bind directly to the surface of the NPs with high affinity, are considered to constitute the hard corona, and proteins that bind to other proteins via weak protein-protein interactions, constitute the soft corona [18]. The corona of various NPs has been studied in detail, and a shared "core"proteome has been established, which comprises lipoproteins, complement and coagulation proteins, and immunoglobulins [19–22]. In the present study we compared our proteomics results to NP protein corona studies.

## Materials and methods

### Preparation of plasma and serum

Blood samples were obtained from healthy, fasting volunteers and collected into ACD-A (Vacuette #455055, Greiner Bio-One, Kremsmünster, Austria), citrate (Vacuette #455322, Greiner Bio-One), serum (BD #369032, BD Biosciences, Franklin Lakes, USA) and EDTA (K2F, BD #368861, BD Biosciences) tubes by a protocol approved by the ethical board of University of Helsinki. Volunteers had not used any medication for the previous 7 days and their informed consent was obtained according to the declaration of Helsinki. Study samples were

collected twice in Helsinki (set I and II) and once in Budapest (set III), the latter to perform fluorescence size exclusion chromatography (Flu-SEC). The average age of the donors in set I was 45 years (range 27–55), 32 years (range 18–51 years) in set II, and 30 years (range 25–36) in set III. Majority of the volunteers were females (80%, 70%, and 67%). Serum, and ACD, citrate and EDTA plasma were obtained during the same blood collection, 2 x 4 mL or 10 mL for each plasma/serum type using 18G needle and tourniquet that was released immediately after the venipuncture, followed by gentle end-over-end mixing 10 times. Blood samples were processed within 30 minutes of sampling to obtain plasma, while serum was allowed to clot for 30 minutes at room temperature (RT) prior centrifugation. Samples for set I and II were centrifuged twice at $2500 \times g$ for 15 minutes at RT (5804R, Eppendorf AG, Hamburg, Germany). For the set III, platelet-free plasma and serum was prepared by centrifuging twice at $2500 \times g$ for 15 minutes at RT (NF 800R, RA 200 swing-out rotor, Nüve, Ankara, Turkey). For the sample sets I and II, equal volumes of individual donors' plasma/serum were combined into a pool of each anticoagulant and serum after the second centrifugation. Pooled plasma/serum samples were analyzed with Coulter T-540 counter (version 3F, Beckman Coulter, Brea, USA) for remnant cell counts, and used for experiments directly without freezing. For the set III, individual donors' samples were kept separate for the Flu-SEC analysis. The details of the workflow are provided in a (S1 Table).

## Isolation of EVs

EVs from set I and II were isolated by ultracentrifugation as follows: pooled plasma (ACD, citrate and EDTA), and serum samples were diluted 1:1 with PBS and centrifuged at 110 000 $\times g_{max}$ for 90 minutes at 4˚C (k-factor 190.2, Type 50.2 Ti, Optima-LE 80K, Beckman Coulter). The supernatant was discarded, and the pellets were resuspended in 20 mL PBS followed by re-centrifugation. The final pellet was suspended in 250 μl PBS, aliquoted to Protein LoBind Tubes (Eppendorf AG) and stored at 4˚C overnight for further analysis, except for TEM and proteomics for which samples were frozen at -80˚C. For the set III, platelet-free plasma and serum were purified by SEC on a 3.5 mL gravity column filled with Sepharose CL-2B gel (GE Healthcare Bio-Sciences AB, Uppsala, Sweden). One mL platelet-free plasma/serum was loaded on the column, the flow through was discarded and purified EVs were eluted with 1 mL PBS buffer.

We have submitted all relevant data of our experiments to the EV-TRACK knowledgebase [23] (http://evtrack.org/, ID: EV190060).

## Flow cytometry analysis

Apogee A50 micro flow cytometer (ApogeeFlow, Hertfordshire, UK) equipped with 405 nm and 488 nm lasers and 525/50 and 575/30 filters was used for measuring EVs. Logarithmic scale and peak height were used. Event level below 100 events/s with 1.5 μL/minute flow rate and 150 mbar pressure was considered acceptable for background. Three washing cycles were performed between the samples. Flow rate was adjusted between 1.5–4.5 μL/minute to keep average event rates below 3000 events/s. All isolated EV samples were diluted 1:20 to obtain ~$1*10^9$ particles/mL and 50 μL was labelled (in technical triplicates) with anti- CD61-PE (clone VI-PL2, BD Biosciences, 1:25) and anti-CD235a-FITC (clone 11E4B-7-6, BD Biosciences, 1:25), matching isotype antibodies at same dilutions, or with Annexin V–Alexa488 (Invitrogen, Carlsbad, USA, 1:20) in the presence of 5 mM CaCl$_2$. Labelling was done for 1 hour at room temperature in the dark. Before analysis, the isolated EV samples were further diluted 1:5. Statistical analysis was conducted using Tukey´s multiple comparison test.

## Size exclusion chromatography with on-line fluorescence detection (Flu-SEC)

Flu-SEC was used to quantify the amount of platelet-derived (CD61+) EVs in serum, and ACD-, citrate- and EDTA-plasma as described earlier [24]. EVs were separated from soluble proteins with the Sepharose CL-2B cross-linked agarose gel (GE Healthcare Bio-Sciences AB). Four μL anti-CD61-PE (clone VI-PL2, 6 μg/mL, BioLegend, USA) was used to label platelet-derived EVs (technical triplicates) in 100 μL of SEC-isolated plasma/serum. Labelling was performed for 30 minutes at 37 ˚C. Ten μL of labeled sample was injected into a Jasco HPLC system (Jasco, Tokyo, Japan) consisting of a PU-2089 pump with a UV-2075 UV/Vis detector and a FP-2020 fluorescence detector controlled by the Chromnav software v. 1.17.02. Tricorn 5/200 glass columns (GE Healthcare Bio-Sciences AB) were packed with Sepharose CL-2B (GE Healthcare Bio-Sciences AB), and the eluent of 1 ml of PBS with a flow rate of 0.5 mL/minute. The fluorescence chromatograms were collected at excitation and emission wavelength corresponding to PE fluorochrome (565/578 nm), and the area under the curve (AUC) of the EV-peak was used to quantify the amount of platelet-derived EVs in the various plasma and serum samples. Statistical analysis was conducted using Tukey´s multiple comparison test.

## Nanoparticle tracking analysis (NTA)

Particle concentration and size distribution were determined with the NTA instrument LM14C (NanoSight LTD., London, UK) equipped with blue (404 nm, 70 mW) laser and sCMOS camera. Settings for data acquisition were: camera level 14, SOP Standard Measurement, autosettings off, polydispersity medium, reproducibility high, and acquisition time 3x 60 sec. The sample dilution was done to 0.1 μm filtered buffer pre-checked for absence of particles. Each sample was diluted to optimal 40–100 particles/frame concentration for measurement. Data were analyzed with NanoSight NTA 3.0 software with the following settings: auto background subtraction/blur/minimum track length, and detection threshold 5.

## Electron microscopy

Negative staining of whole mounted EV samples was performed as described [25]. Briefly, EVs were loaded on 200 mesh grids, fixed with 2% PFA, stained with 2% neutral uranyl acetate, embedded in methyl cellulose uranyl acetate mixture (1.8/0.4%) and viewed with Tecnai 12 (FEI Company, Eindhoven, The Netherland) at 80 kV.

## SDS-PAGE and Western blot analysis

Total of $5{*}10^{10}$ particles of all samples were prepared with 6x Laemmli sample buffer with 10% β-mercaptoethanol and run on 4–20% gradient SDS-PAGE (Mini-Protean TGX Precast protein gel, Bio-Rad, Hercules, USA). For Western blot analysis, proteins were transferred to PVDF membrane (Immobilon-P, Merck Millipore, Burlington, USA) at 200 mA for 90 minutes using wet transfer. Nonspecific binding was blocked by with 3% BSA in Tris-buffered saline containing 0.1% Tween-20 (TBST) for 1 hour followed by primary antibody incubation anti-TSG101 (Clone 51/TSG101 BD Biosciences, 1:500), anti-CD41 (clone SZ22, Beckman Coulter, 1:2000), or anti-CD9 (clone C-4, Santa Cruz Biotechnology, Dallas, USA, 1:500) overnight at 4˚C. The membranes were washed three times with TBST followed by incubation with horseradish peroxidase–conjugated anti-mouse secondary antibody in dilution 1:3000 (GE Healthcare Bio-Sciences AB). After washing, the signal was detected using Bio-Rad Clarity Western ECL Substrate (Bio-Rad) and imaged using LAS3000 imaging system (Fuji, Minato, Japan).

## Mass spectrometry and data analysis

EV pellets corresponding to 100 μg of total protein were adjusted to 25mM Tris, 75mM NaCl; pH 7.5 and applied to 50 μL of albumin depletion resin slurry (Pierce, Thermo Scientific, Rockform, USA) previously washed with same buffer. Flow through and three washes were collected by centrifugation and deemed to be albumin-depleted EV proteins. The protein concentration was measured by bicinchonic acid assay kit (Thermo Fisher Scientific) for the total protein concentration. Equal amount of protein for each sample was dried in speed vacuum (Savant, Thermo Fisher Scientific), and then dissolved in 6 M urea and 100 mM Tris-HCl (pH 7.4). Reconstitution was performed with 10 mM of dithiothreitol (DTT) for 60 minutes at RT, and thereafter 30mM iodoacetamide for 60 minutes in the dark at RT. Protein digestion was performed with trypsin (18 hour at 37˚C) after the consumption of abundant iodoacetamide (30 mM DTT, 60 minutes at RT) and 1:10 dilution with high-purity Milli-Q water (Millipore, Billerica, USA). Finally, the samples were cleaned up with C18 spin columns, and dried in speed vacuum following the dissolution in 0.1% formic acid containing 12.5 fmol Hi3 peptide mixture (Waters Corporation, Beverly, USA) per μl. All procedures were performed according to the manufacturer's instructions. Liquid chromatography: Four microlitre samples corresponding to 1.4 μg of total protein were injected to the ultra-performance liquid chromatography (UPLC) system (Waters Corporation) (Distler et al, 2014). TRIZAIC nanoTile 85 μm x 100mm HSS-T3u wTRAP was applied as a separating device before mass spectrometry (MS). After loading and trapping, the samples were washed for 2 minutes at 8.0 μL/minute with 1% buffer B. The analytical gradient was used as follows: 0–1 minute 1% B; at 2 minutes 5% B; at 65 minutes 30% B; at 78 minutes 50% B; at 80 minutes 85% B; at 83 minutes 85% B; at 84 minutes 1% B; and at 90 minutes 1% B with 450 nL/minute. Buffer A consists of 0.1% formic acid in water and buffer B of 0.1% formic acid in acetonitrile (Sigma-Aldrich). The data was acquired with UDMSE with Synapt G2-S UDMS (Waters Corporation, Billerica, MA) including ion mobility spectroscopy (IMS). The data range was 100–2000m/z, scan time 1 s, IMS wave velocity 650ms[1]. Calibration was performed by Glu1- fibrinopeptide B MS2 fragments and Glu1- fibrinopeptide B precursor ion, used during the acquisitions as a lock mass. Progenesis QI for proteomics software (Version 3, Nonlinear Dynamics) was used for processing of raw files. Peptide identification was run with Uniprot human FASTA sequences (UniprotKB Release 2015_09, 20205 sequence entries), and label-free protein quantification performed with Hi-N method (Protein Lynx Global Server) (Silva et al, 2006). The samples were spiked with 12.5 fmol/μL of CLPB_ECOLI (P63285, ClpB protein) peptides (Hi3 E. Coli Standard, Waters). Digesting reagent was trypsin and 2 miss cleavages were allowed. The peptide identification parameters were: fixed modification of cysteine (carbamidomethyl) and variable modification of methionine (oxidation). The peptide error tolerance was set to maximally 10 ppm and the false discovery rate limited to less than 1% and default values (in Progenesis QI for Proteomics) were used for the rest of the parameters. The quantified proteins in all comparisons were compared by ANOVA on a protein-to-protein basis and their expression levels were considered significantly different, if ANOVA p-value was less than 0.05. Principal component analysis (PCA), offering the visualization of the main axes of variation in the data groups, was performed by Progenesis QI for proteomics.

## Statistics

Statistical analysis of the flu-SEC experiment was performed by using Prism 8 (GraphPad Software, LLC, USA). One-way ANOVA test was performed without pairing and assuming Gaussian distribution and equal variance followed by Tukey multiple comparisons test with 95%

confidence interval. Saphiro-Wilk test was used to test normal distribution, while the Brown–Forsythe test was used for the equality of group variances. Statistical analysis of flow cytometry and proteomics experiments was performed with IBM SPSS version 24 (IBM Corp., Armonk, USA, Released 2016). The data was tested for normality using Saphiro-Wilk test, after which Kruskall-Wallis non-parametric test was used followed by post-hoc Dunn test. Pearson correlation test was used to examine whether there were correlations between proteins analyzed by proteomics. The level of significance was set at $p \leq 0.05$ in all experiments.

### Data repository

The raw files were converted with MSConvert (ProteoWizard) to mzML-files. The mass spectrometry proteomics data have been deposited to the ProteomeXchange Consortium via the PRIDE (Vizcaino et al 2014) partner repository with the dataset identifier PXD015375.

## Results

### General characterization of EVs derived from ACD-, citrate- and EDTA-anticoagulated plasma and serum

In this study, we investigated how different anticoagulants and clotting affect the characteristics of blood-derived EVs. To avoid contamination with any residual platelets and/or residual platelets generating artificial EVs, all samples were centrifuged within 30 minutes after blood collection at 2500 × g 15 minutes (twice) as recommended previously [4, 6, 26]. Little or no residual platelets were detected in the samples (S1A Fig). Particle concentrations and size distributions were determined with NTA, and the vesicle morphology in samples was confirmed with TEM. The concentrations of particles calculated per original sample volume of the different plasmas or serum were not statistically different in blood collection set I ($p = 0.631$), but in set II, some variation was found in EV concentrations between serum and anticoagulated samples ($p = 0.001$; S1B Fig). Statistical testing for size distribution of EVs was performed for set II which had three technical replicates, but essentially similar data was also obtained from samples of set I with duplicate samples. The analysis revealed statistically significant differences when measured by bins of size classes. The significant increase of particle counts was seen with serum-derived EVs that differed from ACD, citrate and EDTA plasma–derived EVs in size classes 0–100 nm ($p = 0.002$ (ACD), $p = 0.001$ (citrate), $p = 0.023$ (EDTA) and 101–200 nm $p = 0.044$ (ACD) (S1C Fig). By TEM, all EVs irrespective of plasma or serum source had a similar morphology typically attributed to EVs (S1D Fig). To investigate the protein content of EVs, equal particle numbers based on NTA were loaded into SDS-PAGE gels. The overall protein patterns were similar in the ACD- and citrate-derived EV samples, whereas the protein bands from EDTA- and serum-derived EVs showed visible differences (S1E Fig). In contrast, typical EV-marker proteins as TSG101 and CD9 were detected at comparable levels when analyzed by Western blotting (S1F Fig).

### Serum and citrate plasma contain increased numbers of platelet-derived EVs compared to ACD and EDTA plasma samples

In a seminal study of George et al. [7], serum was shown to contain more microparticles i.e. large EVs than ACD plasma. This prompted us to investigate this claim anew with contemporary analytical methods: EV-dedicated flow cytometry (Apogee 50), flu-SEC and Western blotting. EVs of different plasma and serum samples isolated by ultracentrifugation and normalized by particle numbers based on NTA, were analyzed with flow cytometry by anti-CD61 and -CD235a labelling to compare platelet and erythrocyte derived EV proportions in

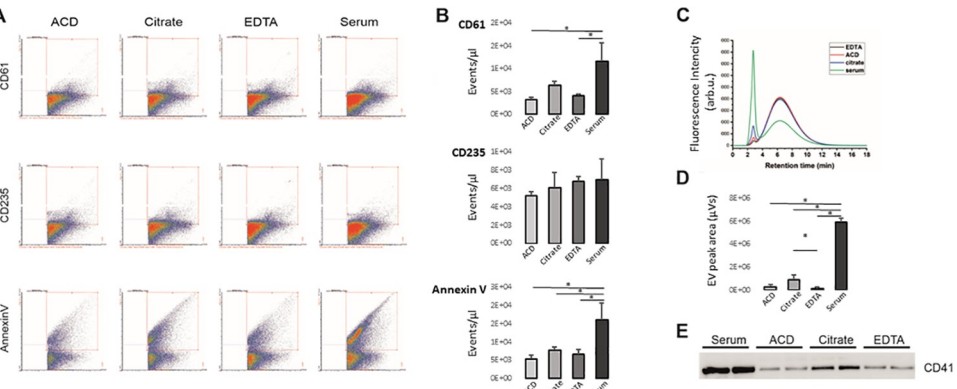

**Fig 1. Flow cytometry, fluorescence size exclusion chromatography, and Western blot of extracellular vesicles isolated from ACD-, citrate-, and EDTA-plasma and serum samples.** A) Representative images of the Apogee flow cytometry analysis with anti-CD61-PE, anti-CD235-FITC and annexin V-FITC, respectively, B)Box blots show the average of events/µl (n = 3) for the labelled EVs isolated from plasma and serum samples. For the platelet marker CD61, significant increases in events were detected for serum and citrate-anticoagulated plasma EVs. In contrast, similar concentrations of erythrocyte (CD235a) EVs were present in plasma and serum. Serum EVs bound more a Annexin V per EVs compared to plasma EVs, C) representative chromatograms of plasma- and serum-derived EVs labelled with anti-CD61-PE and characterized with fluorescent size exclusion chromatography (n = 3). EVs were eluted with the void volume of the column at 2.8 min retention time, and the peak at 6.35 min retention time corresponded to the unbound antibody. D) The area under the curve values of the first peak were correlated with the amount of CD61+ EVs in plasma and serum samples. E) probing with anti-CD41 of the Western blots of the plasma and serum -derived EVs to detect platelet-derived EVs (TSG101 and CD9 are shown in S1 Fig). Each lane was loaded with a comparable number of particles based on NTA. Asterisks (*) indicates $p < 0.05$.

plasma and serum, respectively. There were less CD235a+ EVs compared to CD61+ EVs in all the samples irrespective of the anticoagulant used (Fig 1A). In contrast, more CD61+ events were detected in the serum samples compared to the ACD ($p = 0.006$) and EDTA ($p = 0.001$) samples, and although there was a similar trend with the citrate sample, it was not statistically significant. Statistical testing was performed for set II which had three technical replicates, but essentially similar data was also obtained from samples of set I with duplicate samples. Annexin V binding to EVs was analyzed to yield information about phosphatidylserine (PS) expression relevant for procoagulant activity. Staining with (fluorescently-labelled) annexin V was low in all plasma-derived EV samples compared to serum EVs (Fig 1A). Statistical analysis showed significant differences (S2 Table) between serum and ACD, citrate, and EDTA EVs ($p = 0.003$, $p = 0.013$, $p = 0.006$, respectively).

In sample set III, the plasma and serum samples were labelled with anti-CD61 followed by flu-SEC (Fig 1B). Comparison of the chromatograms of the plasma samples with that of the serum sample demonstrated that the concentration of CD61+ EVs was significantly increased in the serum samples compared to ACD, EDTA and citrate plasma (all $p<0.001$; S3 Table), even up to 40-fold compared to EDTA. While there was no difference in the concentration of CD61+ EVs between the EDTA- and ACD plasma samples, citrate-anticoagulated plasma also had an increased number of CD61+ EVs compared to EDTA (p = 0.046).

When equal particle numbers were loaded on Western blot and probed with anti-CD41 to detect the platelet glycoprotein IIb, the strongest CD41 signal was in the serum-derived EVs, followed by the EVs from citrate plasma, whereas in the ACD- and EDTA-derived EVs the band intensities were equal (Fig 1C). The Western blot, Apogee flow cytometry and Flu-SEC results confirmed a high concentration of platelet-derived EVs in serum and citrate plasma compared to the ACD- and EDTA-plasma samples. Finally, also proteomics confirmed an increase in the presence of CD41 protein in serum and EDTA plasma–derived EV samples

compared to ACD and citrate plasma -derived EVs (p = 0.04) in the quantitative proteomics of albumin-depleted samples (Fig 1D).

## Protein cargo of the EV corona is modified by anticoagulation or its absence

Intrigued by the observations of the different protein bands in the SDS gel of the isolated EVs (S1E Fig), we performed a quantitative proteomic analysis for both albumin-depleted EVs and non-albumin-depleted EVs isolated from ACD-, citrate-, and EDTA-plasma and serum samples. In total, 225 proteins were detected from the albumin-depleted and 165 from the non-depleted EV samples with mass spectrometry Since all identified and quantified proteins (S4 Table) from the non-albumin depleted samples (n = 91 out of 165) were present in the list of albumin-depleted samples (n = 138 out of 225), we focused on the albumin-depleted proteome. Abundances of the EV proteins were used to perform principal component analysis (PCA). PCA is available in the Progenesis QI for proteomics—software package. It is an unsupervised method to determine relations between the samples and here we directly used the imported mass spectrometry data of the reliable proteins identified with two or more unique peptides. The principal axes of abundance variation are calculated by the software from protein abundances in the samples. When two or more unique peptide -containing proteins with $p \leq 0.05$ ANOVA were plotted in PCA, there was a complete separation of the samples into four separate groups according to anticoagulation/coagulation (Fig 2). The Pearson's correlation coefficient was calculated for the 138 proteins. A threshold for the p-value was set to 0.05 to define statistical significance. More than 80% of protein pairs were positively or negatively correlated (S5 Table).

The differences in the EV proteomes from plasma and serum samples were quantitative. A gene ontology analysis was conducted with the FunRich analysis tool [27] using human genome as background. Of the 138 proteins, 107 were mapped for molecular function, cellular component and biological process GO terms. The GO terms with $p$-value $\leq 0.05$ (Holm-Bonferroni corrected) are shown in Fig 3, and a detailed list of the mapped genes is presented in S6 Table. The most prominent cellular components in GO terms (Fig 3A) were exosomes (GO:0070062), cytoplasm (GO:0005737) and extracellular (GO:0005576) comprising 77.7%, 68.0% and 66.0% of the identified proteins, respectively. The enriched biological process GO terms (Fig 3B) were cell growth and/or maintenance (GO:0016049) and immune response (GO:0006955), comprising 32.4% and 22.5%, respectively. For the molecular function (Fig 3C)

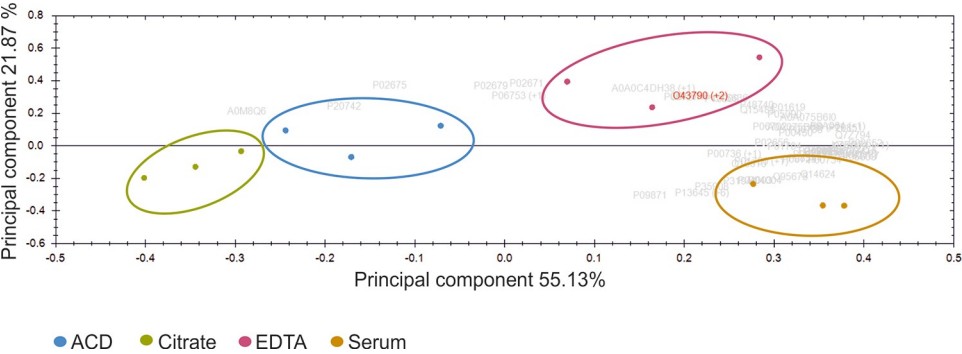

**Fig 2. Principal component analysis (PCA) of protein quantification data with two or more unique peptide -containing proteins with ANOVA significance p≤0.05.** Green: citrate; blue: ACD; red: EDTA; yellow: serum. Each sample was run in triplicates.

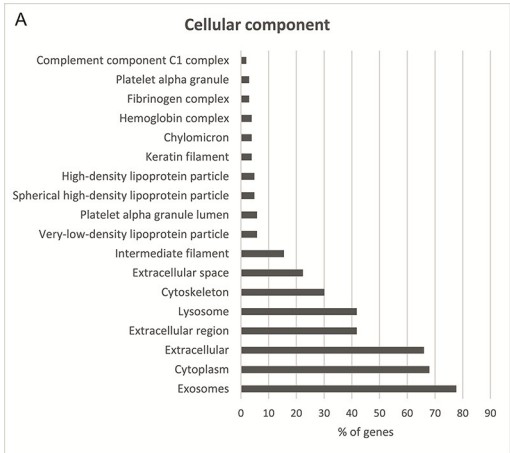

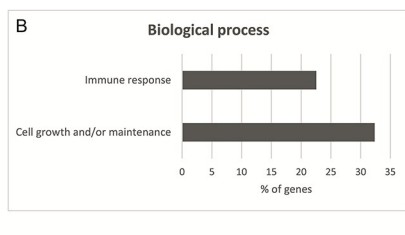

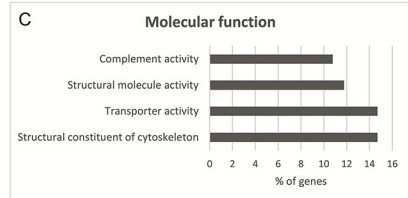

**Fig 3. Gene ontology enrichment analysis of identified proteins from EVs with two or more unique peptide–containing proteins and p-value ≤0.05 (Holm-Bonferroni corrected) performed with FunRich 2.1.2 software using human genome as a background.** A) Cellular component enrichment analysis, B) molecular function enrichment analysis, and C) biological process enrichment analysis.

enriched GO terms were transporter activity (GO:0005215), structural constituent of cytoskeleton (GO:0005200), structural molecule activity (GO:0005198), and complement activity (GO:0006956) comprising 14.7%, 14.7%, 11.8%, and 10.8% of the identified proteins, respectively.

Of the 138 proteins, 35 showed significant differences in concentration (ANOVA with Kruskall-Wallis test followed by Dunn´s test), between the plasma and serum EV samples (Table 1). The majority of these proteins [21] were upregulated in the EVs isolated from serum, 12 proteins in the EVs isolated from EDTA-plasma, and two in the EVs isolated from ACD-plasma. The further analysis of 21 proteins with highest mean in serum EVs revealed enriched molecular function in transporter activity (GO:0005215), including apolipoproteins A1, A2 and C3, transferrin and vitamin D binding protein. Additionally, the EVs displayed differences in the amounts of coagulation and complement proteins, protease inhibitors and different classes of immunoglobulins (Table 1). The most prominent GO term in molecular function of combined EDTA and ACD plasma EVs was protein binding (GO:0005515).

The proteome of plasma EVs has been reported in several studies and can be obtained from Exocarta [28] and EVpedia [29]. To understand our results, we performed data comparisons between our results and the results from two other studies which we deemed most relevant by their study design. The study of Kalra et al. compared the proteome of plasma EVs which had been isolated with three different isolation methods: differential ultracentrifugation (dUC), immunoaffinity, and discontinuous gradient ultracentrifugation (DG) [30]. This study reported 56 common proteins with our proteome from dUC and DG samples, including complement proteins, immunoglobulins, apolipoproteins and keratins. Similar results have also been reported in studies using dUC [31–34] or SEC as isolation method [35–37]. In the study of Karimi et al. it was shown that by combining dUC, DG and SEC, it is possible to avoid lipoprotein and soluble protein contamination and to isolate highly purified plasma EVs for proteomic analysis [36]. Comparing our 138 proteins from UC-isolated EVs with those of dUC-isolated EVs [30] and dUC combined with DG and followed by SEC isolated EVs [36] showed that 46 proteins were shared with both of the other data sets, 9 proteins only with Kalra et al. and 53 proteins only with Karimi et al. (Fig 4).

**Table 1. Proteins (n = 35) showing significant difference between the plasma and serum EV samples (Anova with Kruskall-Wallis test followed by Dunn´s test, *p*-value ≤ 0.05) are shown.** Five keratins listed here were not excluded although they are considered to originate from exogenous sources such as skin or hair.

| | Group | Mean | SD | ACD *p*-value | Citrate *p*-value | EDTA *p*-value |
|---|---|---|---|---|---|---|
| Alpha-1-antitrypsin | Citrate | 21128.5 | 3673.3 | 0.174 | | |
| | EDTA | 57249.3 | 23861.6 | 0.571 | 0.054 | |
| | Serum | 84532.5 | 5897.2 | 0.141 | 0.005* | 0.365 |
| APOC4-APOC2 readthrough (NMD candidate) | Citrate | 2899.5 | 269.1 | 0.308 | | |
| | EDTA | 14746.6 | 3666.4 | 0.308 | 0.042* | |
| | Serum | 23054.3 | 322.9 | 0.042* | 0.002* | 0.308 |
| Apolipoprotein A-I | Citrate | 327405.2 | 26276.5 | 0.571 | | |
| | EDTA | 763780.0 | 136205.2 | 0.174 | 0.054 | |
| | Serum | 964279.7 | 111475.8 | 0.031* | 0.007* | 0.428 |
| Apolipoprotein A-II | Citrate | 9432.0 | 1918.8 | 0.428 | | |
| | EDTA | 32012.0 | 10122.1 | 0.213 | 0.042* | |
| | Serum | 46145.7 | 3288.2 | 0.042* | 0.005* | 0.428 |
| Apolipoprotein C-III | Citrate | 12313.4 | 2369.5 | 0.734 | | |
| | EDTA | 45020.7 | 29899.0 | 0.089 | 0.042* | |
| | Serum | 40119.1 | 2971.9 | 0.042* | 0.017* | 0.734 |
| C4b-binding protein beta chain | Citrate | 32830.5 | 4683.5 | 0.308 | | |
| | EDTA | 145233.1 | 92104.5 | 0.174 | 0.017* | |
| | Serum | 129147.1 | 21501.1 | 0.089 | 0.007* | 0.734 |
| Clusterin | Citrate | 15921.9 | 2556.5 | 0.174 | | |
| | EDTA | 64159.9 | 2776.6 | 0.113 | 0.003* | |
| | Serum | 26015. | 4840.8 | 1.00 | 0.174 | 0.113 |
| Complement C1s subcomponent | Citrate | 130601.2 | 18724.9 | 0.308 | | |
| | EDTA | 77030.6 | 18048.8 | 0.042* | 0.308 | |
| | Serum | 574764.3 | 80460.7 | 0.308 | 0.042* | 0.002* |
| Complement factor H | Citrate | 37145.7 | 4413.8 | 0.571 | | |
| | EDTA | 97696.5 | 24032.5 | 0.213 | 0.07 | |
| | Serum | 161619.3 | 19798.8 | 0.024* | 0.005* | 0.308 |
| Extracellular matrix protein 1 | Citrate | 1361.0 | 349.4 | 0.910 | | |
| | EDTA | 2848.3 | 764.6 | 0.113 | 0.141 | |
| | Serum | 7871.9 | 5854.9 | 0.009* | 0.013* | 0.3008 |
| Far upstream element-binding protein 2 | Citrate | 40731.4 | 11563.2 | 0.428 | | |
| | EDTA | 130437.5 | 10503.6 | 0.428 | 0.113 | |
| | Serum | 272062.8 | 40803.3 | 0.042* | 0.005* | 0.213 |
| Fibrinogen alpha chain | Citrate | 286470.0 | 80015.7 | 0.497 | | |
| | EDTA | 1040565.0 | 140516.8 | 0.213 | 0.054 | |
| | Serum | 178365.8 | 33284.6 | 0.089 | 0.308 | 0.003* |
| Fibrinogen beta chain | Citrate | 247084.9 | 82918.2 | 0.571 | | |
| | EDTA | 796061.9 | 71036.9 | 0.213 | 0.07 | |
| | Serum | 52282.2 | 24865.0 | 0.07 | 0.213 | 0.002* |
| Fibrinogen gamma chain | Citrate | 214132.6 | 59581.7 | 0.428 | | |
| | EDTA | 681749.2 | 115693.1 | 0.428 | 0.113 | |
| | Serum | 112276.9 | 6000.7 | 0.042* | 0.213 | 0.005* |

(*Continued*)

**Table 1.** (Continued)

| | Group | Mean | SD | ACD *p*-value | Citrate *p*-value | EDTA *p*-value |
|---|---|---|---|---|---|---|
| | | | | **Kruskall-Wallis** | | |
| Ficolin-3 | Citrate | 39691.6 | 6389.57 | 0.571 | | |
| | EDTA | 3262480.8 | 16914469 | 0.024* | 0.005* | |
| | Serum | 175056.8 | 89930.4 | 0.213 | 0.07 | 0.308 |
| Haptoglobin-related protein | Citrate | 72310.9 | 6719.7 | 0.005* | | |
| | EDTA | 216281.3 | 208441.5 | 0.308 | 0.07 | |
| | Serum | 121909.9 | 24163.3 | 0.174 | 0.141 | 0.734 |
| Immunoglobulin heavy variable 5–51 | Citrate | 2005.0 | 146.1 | 0.089 | | |
| | EDTA | 27441.8 | 12458.0 | 0.174 | 0.002* | |
| | Serum | 3673.2 | 79.9 | 0.734 | 0.174 | 0.089 |
| Immunoglobulin kappa variable 3–20 | Citrate | 23823.5 | 1779.9 | 0.571 | | |
| | EDTA | 79664.0 | 33562.0 | 0.024* | 0.005* | |
| | Serum | 46655.7 | 3347.3 | 0.213 | 0.07 | 0.308 |
| Immunoglobulin lambda constant 2 | Citrate | 34925.7 | 9817.4 | 0.365 | | |
| | EDTA | 90591.5 | 19342.8 | 0.365 | 0.07 | |
| | Serum | 145174.1 | 28041.9 | 0.042* | 0.003* | 0.258 |
| Immunoglobulin lambda constant 7 | Citrate | 3043.6 | 1419.7 | 0.428 | | |
| | EDTA | 1933.9 | 501.0 | 0.213 | 0.651 | |
| | Serum | 438.4 | 349.6 | 0.007* | 0.054 | 0.141 |
| Immunoglobulin lambda variable 2–14 | Citrate | 11752.6 | 3612.3 | 0.910 | | |
| | EDTA | 21819.8 | 4915.8 | 0.054 | 0.07 | |
| | Serum | 23822.8 | 8356.2 | 0.024* | 0.031* | 0.734 |
| Immunoglobulin lambda-like polypeptide 5 | Citrate | 67358.0 | 15634.6 | 0.428 | | |
| | EDTA | 186777.2 | 68310.8 | 0.113 | 0.017* | |
| | Serum | 164633.0 | 4839.5 | 0.089 | 0.013* | 0.91 |
| Inter-alpha-trypsin inhibitor heavy chain H4 | Citrate | 2136.6 | 130.5 | 0.910 | | |
| | EDTA | 4742.3 | 2796.6 | 0.141 | 0.113 | |
| | Serum | 27310.24 | 1537.1 | 0.013* | 0.009* | 0.308 |
| Keratin type II cuticular Hb6 | Citrate | 4676.76 | 1131.9 | 0.571 | | |
| | EDTA | 226331.9 | 72476.3 | 0.005* | 0.024* | |
| | Serum | 9422.9 | 332.7 | 0.07 | 0.231 | 0.308 |
| Keratin type II cytoskeletal 1b | Citrate | 6686.7 | 1671.8 | 0.571 | | |
| | EDTA | 23137.7 | 6736.6 | 0.089 | 0.024* | |
| | Serum | 25899.7 | 3549.7 | 0.07 | 0.017* | 0.91 |
| Keratin type II cytoskeletal 3 | Citrate | 8968.3 | 1474.3 | 1.00 | | |
| | EDTA | 16936.0 | 5221.0 | 0.571 | 0.089 | |
| | Serum | 25805.0 | 3225.4 | 0.089 | 0.005* | 0.258 |
| Keratin type II cytoskeletal 7 | Citrate | 9786.3 | 5026.1 | 0.428 | | |
| | EDTA | 28739.1 | 9806.1 | 0.428 | 0.113 | |
| | Serum | 78335.4 | 27832.8 | 0.042* | 0.005* | 0.213 |
| Keratin type II cytoskeletal 75 | Citrate | 17435.1 | 3417.5 | 0.910 | | |
| | EDTA | 23954.2 | 5558.9 | 0.231 | 0.258 | |
| | Serum | 39640.8 | 7021.8 | 0.013* | 0.017* | 0.213 |
| Mannan-binding lectin serine protease 1 | Citrate | 19359.2 | 838.0 | 1.00 | | |
| | EDTA | 369877.9 | 451573.1 | 0.089 | 0.005* | |
| | Serum | 82930.6 | 47213.7 | 0.308 | 0.031* | 0.497 |

(*Continued*)

**Table 1.** (Continued)

| | | Kruskall-Wallis | | | | |
|---|---|---|---|---|---|---|
| | Group | Mean | SD | ACD | Citrate | EDTA |
| | | | | *p*-value | *p*-value | *p*-value |
| NFX1-type zinc finger-containing protein 1 | Citrate | 1665.6 | 406.6 | 0.258 | | |
| | EDTA | 5768.4 | 2119.3 | 0.365 | 0.042* | |
| | Serum | 9485.0 | 2649.1 | 0.07 | 0.003* | 0.365 |
| Pro-thrombin | Citrate | 16532.6 | 2078.7 | 0.174 | | |
| | EDTA | 36920.9 | 16154.9 | 1.00 | 0.174 | |
| | Serum | 92027.3 | 11207.8 | 0.113 | 0.003* | 0.113 |
| Putative uncharacterized protein MYH16 | Citrate | 393.4 | 93.2 | 0.734 | | |
| | EDTA | 5638.6 | 2475.0 | 0.024* | 0.009* | |
| | Serum | 734.2 | 125.3 | 0.308 | 0.174 | 0.213 |
| Serotransferrin | Citrate | 246039.8 | 45043.8 | 0.428 | | |
| | EDTA | 603798.7 | 62743.3 | 0.258 | 0.054 | |
| | Serum | 1127260.8 | 200381.7 | 0.031* | 0.003* | 0.308* |
| Transthyretin | Citrate | 850.2 | 268.5 | 0.024* | | |
| | EDTA | 5663.8 | 5519.4 | 1.00 | 0.024* | |
| | Serum | 1237.8 | 259.7 | 0.113 | 0.497 | 0.113 |
| Vitamin D-binding protein | Citrate | 11973.0 | 1888.2 | 0.497 | | |
| | EDTA | 43094.2 | 16511.6 | 0.258 | 0.07 | |
| | Serum | 79524.5 | 15148.7 | 0.042* | 0.007* | 0.365 |

Because we found similarities in the EV proteome of our study with these two previous studies, which had used different and more diligent EV isolation procedures, we hypothesized that there may be a shared "EV protein corona". To investigate this hypothesis, we compared

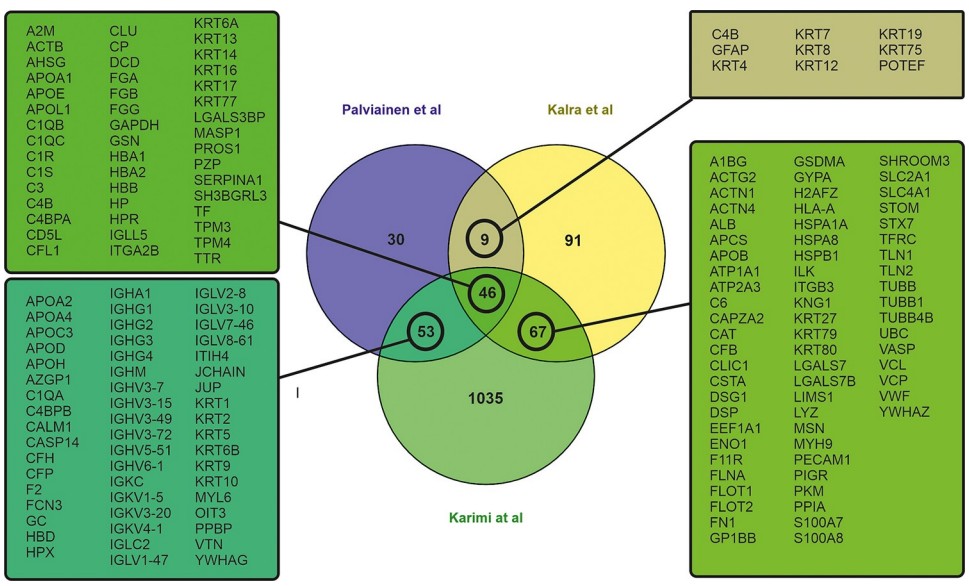

**Fig 4. Venn diagram of overlapping proteins from a comparison of the proteomic data of our study and two previous studies of proteome of EVs [30, 36].**

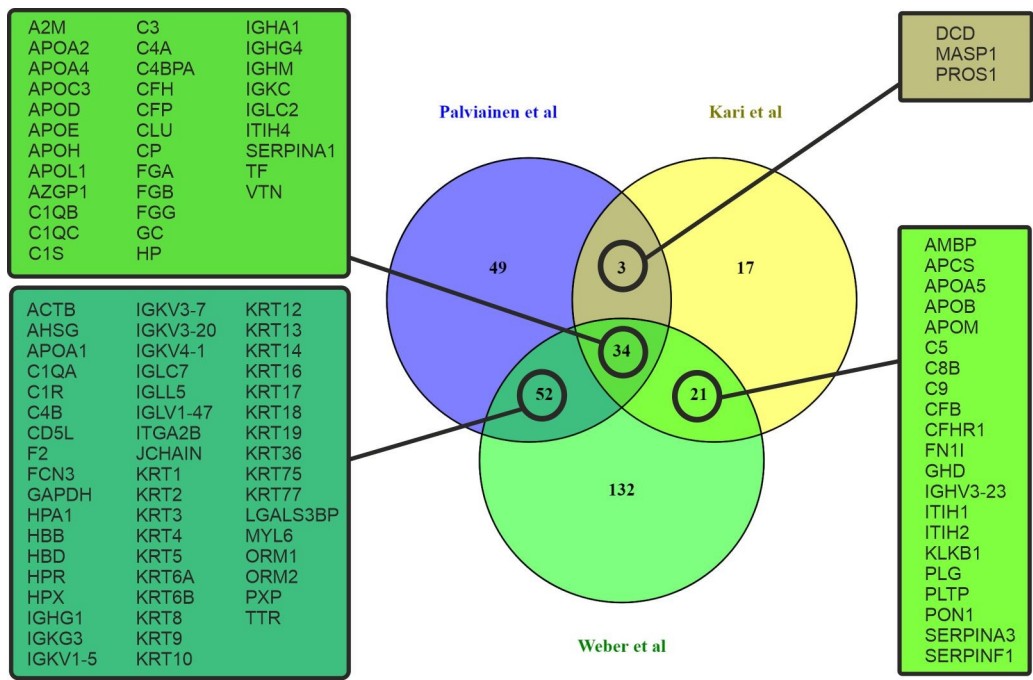

**Fig 5. Venn diagram comparing the proteomic data of our study to a shared proteome from two previous studies of nanoparticle corona proteins accumulated from plasma [22, 38] showing a set of overlapping proteins, which suggests that a common corona proteome of plasma proteins accumulates on EVs and synthetic particles [22, 38].**

our EV proteome to that of NPs, which had been incubated with plasma to generate a protein corona. We compared the list of the 108 proteins which were shared with our study and two previous EV studies, assuming that these proteins belonged to the EV protein corona, to a set of 84 proteins which were shared between two previous NP proteomic studies [22, 38]. Kari et al. used multi-parametric surface plasmon resonance to identify hard and soft protein corona of liposomes *in situ* [38], and Weber et al. analyzed the polystyrene NP protein corona after isolation by dUC and asymmetric flow field-flow fractionation [22]. In total, 89 of the "EV corona proteins" were identical to those reported to belong to the NP corona proteins (65%; Fig 5). A gene ontology analysis with the FunRich analysis tool [27] using human genome as background enabled us to map 73 proteins (out of the 89 common proteins) for molecular function, cellular component and biological process GO terms (S6 Table). The most prominent cellular component GO terms were exosomes (GO:0070062), extracellular (GO:0005576) and cytoplasm (GO: 0005737) comprising 86%, 75% and 63% of the identified proteins, respectively. The enriched biological process GO terms were cell growth and/or maintenance (GO:0016049), immune response (GO:0006955) and transport (GO: 0006810) comprising 32%, 30% and 20%, respectively. For the molecular function enriched GO terms were transporter activity (GO:0005215), structural constituent of cytoskeleton (GO:0005200) and complement activity (GO: 0006956), comprising 20%, 17% and 14% of the identified proteins, respectively.

## Discussion

At present, blood-derived EVs are a potential source for new biomarkers. Since a blood sample is an *ex vivo* sample, the impact of the pre-analytical choices prior to EV isolation needs to be

accounted for to make intelligent use of high-throughput data utilizing samples of large cohorts and biobanks. Therefore, we compared EVs isolated from ACD-, citrate and EDTA-plasma, and serum, to investigate the effects of anticoagulation on the cellular origin and protein composition of the isolated EVs. An early study by George et al. compared large EVs from plasma and serum, and they showed that serum contained more platelet-derived microparticles than ACD plasma [7]. Because their study disregarded a large bulk of EVs, we readdressed this issue with contemporary analytical EV methodologies. We showed with four different methods that the serum samples indeed contain a significantly higher concentration of platelet-derived EVs than the plasma samples, and that the serum-derived EVs bound more Annexin V. The number of platelet-derived EVs was also increased in citrate plasma compared to ACD and EDTA, which confirms previous results [39]. This suggests that citrate inhibits platelet activation less efficiently than ACD and EDTA, which results in generation platelet-derived EVs in blood samples during and after blood collection, especially if there is a delay between the sample collection and processing. This is significant, because uncontrolled release of platelet-derived EVs affects biomarker discovery, e.g., platelet-derived EVs were shown to skew the miRNA populations in patient and control samples [40]. Although the total concentration and size distribution of the plasma-derived EVs were essentially similar between the various anticoagulated plasma and serum samples, a slight difference in the particle numbers was observed between sample sets I and II, where in set II, the concentration of serum-derived EVs was higher than in plasma. This difference may be explained by the variation between the donor pools, and one of the limitations of this study was that the EV analyses could not be associated to the individual donor characteristics such as age or gender So far, very few studies have addressed how such factors impact the EV concentrations in healthy populations [41–43]. However, the use of large, well-prepared sample pools enabled us to apply the same material systematically in multiple analyses, strengthening the comparison of the results between analytics.

Ultracentrifugation is still the most used method for EV isolation from plasma (>60%) or serum (>55%) reported in EV-TRACK (http://evtrack.org/, accessed 20.01.2020), although other methods e.g. SEC and density gradient ultracentrifugation have been adopted. Thus, to mimic the situation with most studies, the isolation of EVs in this study was performed with dUC with one washing step, except Flu-SEC. The use of dUC is another limitation of this study, because dUC is known to co-isolate protein aggregates and lipoproteins [44, 45]. Additionally, the recovery of EVs with dUC is variable (10–80%) and often low [46] and it excludes the smallest of EVs and exomeres, to which little attention has been focused so far [47, 48]. Thus, this study did not address the possible alterations in the smallest EV subpopulations.

Several studies of proteomics on plasma and/or serum-derived EVs using various isolation methods have been published [32, 34–37], and regardless of the isolation method, the published EV proteomes show a remarkable similarity. This indicates that certain plasma proteins are tightly bound to the EV surface constituting a true protein corona rather than being mere co-isolating contaminants. The proteome of EVs comprised lipoproteins, immunoglobulins, and coagulation and complement proteins. The presence of several transporter proteins in the protein corona suggests that EV surfaces may have a previously unconsidered role in biomolecule transportation. As the enzymatic removal of the EV surface proteins has been shown to hamper cellular uptake [49, 50], the protein corona may affect the functional properties of EVs. The heterogeneity of the protein corona was shown to affect the hydrodynamic range of EVs, as well as their mobility in solution and extracellular matrix [51].

Our proteomics analysis of the albumin-depleted EVs from plasma and serum showed an overlap not only with the previous EV studies [30, 36], but also with NP studies [22, 38]. The NP studies chosen for comparison applied isolation methods asymmetric field flow

fractionation and multi-parametric surface plasmon resonance which are considered not to co-isolate soluble proteins, and thereby strengthen the concept that the shared proteins with EVs are unlikely to be co-isolated—artefacts. Of note, in some corona studies NPs added to plasma were isolated by methods which are also used to isolate EVs from plasma and serum [52–56]. In such studies, one cannot conclude whether the reported protein corona of NPs originates truly from NPs, from co-isolated EVs, or both. In future studies, this aspect should be taken into account when conducting NP protein corona research, as also noted by a recent viewpoint article [57].

The similarities between the shared extravesicular cargo of EVs and NPs imply that both EVs and NPs accumulate a selected plasma protein corona on their surface, which may be applicable to biomarker discovery. Many of the surface-attached proteins of EVs associate with pathological states and thus could be useful for diagnostics. For example, synovial fluid-derived EVs of rheumatic arthritis patients were shown to carry complement factors and complement activator molecules [58], and the adhered protein composition of EVs was shown to reflect the disease state in preeclampsia [59].

To our knowledge, this is the first study highlighting the importance of the EV corona. To conclude, we state that plasma obtained with different anticoagulant and/or serum should not be mixed as a source of EVs for biomarker discovery. This is particularly relevant for clinical studies and biobank samples. Secondly, an enrichment of platelet-derived EVs from serum samples may offer new opportunities to study platelet-associated pathologies. Thirdly, the comparison of proteomic studies suggests that EVs have a protein corona that includes both soft and hard coronas offering an interactive surface with biological membranes, which may offer novel insights in the pathophysiological roles of EVs. Since the investigations on the communicasome role of EVs have so far predominantly concentrated on the intravesicular cargo and regarded the corona proteins simply as contaminants, we suggest that this "piggyback" cargo should be re-evaluated. We expect our study to pave a way for more discoveries regarding the role of the extracellular cargo of EVs.

## Supporting information

**S1 Raw images.**
(PDF)

**S1 Fig. General characterization of plasma and serum–derived EVs.**
(JPG)

**S1 Table. Workflow of the study.**
(DOCX)

**S2 Table. Flow cytometry analysis of plasma and serum–derived EVs.**
(DOCX)

**S3 Table. Flu-SEC analysis of plasma and serum–derived EVs.**
(DOCX)

**S4 Table. Identified and quantified proteins.**
(XLSX)

**S5 Table. Pearson correlation matrix of identified and quantified proteins.**
(XLSX)

**S6 Table. A gene ontology analysis of "EV corona proteins".**
(DOCX)

**S7 Table. A gene ontology analysis of "NP and EV corona proteins".**
(DOCX)

## Acknowledgments

We thank all the volunteers for participating in this study.

## Author Contributions

**Conceptualization:** Maarit Takatalo, Pia R. M. Siljander.

**Data curation:** Sakari Joenväärä.

**Formal analysis:** Mari Palviainen, Zoltán Varga, Diána Kitka.

**Funding acquisition:** Risto Renkonen, Pia R. M. Siljander.

**Investigation:** Mari Palviainen, Mayank Saraswat, Diána Kitka, Maarit Neuvonen, Maija Puhka, Sakari Joenväärä, Maarit Takatalo.

**Methodology:** Mari Palviainen, Mayank Saraswat, Zoltán Varga, Maarit Takatalo, Pia R. M. Siljander.

**Resources:** Risto Renkonen.

**Supervision:** Pia R. M. Siljander.

**Validation:** Zoltán Varga, Diána Kitka.

**Visualization:** Mari Palviainen.

**Writing – original draft:** Mari Palviainen, Mayank Saraswat, Maarit Takatalo.

**Writing – review & editing:** Zoltán Varga, Rienk Nieuwland, Pia R. M. Siljander.

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
