## [Decision Letter · Decision Letter 0]

29 Apr 2020

PONE-D-20-08075

Extracellular vesicles from human plasma and serum are carriers of extravesicular cargo ─ implications for biomarker discovery

PLOS ONE

Dear Dr Siljander,

Thank you for submitting your manuscript to PLOS ONE. After careful consideration, we feel that it has merit but does not fully meet PLOS ONE’s publication criteria as it currently stands. Therefore, we invite you to submit a revised version of the manuscript that addresses the points raised during the review process.

The reviewers were both very favorable in their reviews, but noted some changes that need to be made to the manuscript.  Because this journal focuses on transparency of methods and clarity of presentation, please carefully address the comments requesting explanation of the statistical approaches.  Also, please check on the figure quality to make sure that they will have sufficient resolution for publication.

We would appreciate receiving your revised manuscript by Jun 13 2020 11:59PM. To enhance the reproducibility of your results, we recommend that if applicable you deposit your laboratory protocols in protocols.io, where a protocol can be assigned its own identifier (DOI) such that it can be cited independently in the future. For instructions see: http://journals.plos.org/plosone/s/submission-guidelines#loc-laboratory-protocols

We look forward to receiving your revised manuscript.

Kind regards,

John Matthew Koomen, PhD

Academic Editor

PLOS ONE

Journal Requirements:

4. Thank you for including your ethic statement in the methods section:

"Blood samples were obtained from healthy, fasting volunteers and collected into ACD-A (Vacuette #455055, Greiner Bio-One, Kremsmünster, Austria), citrate (Vacuette #455322, Greiner Bio-One), serum (BD #369032, BD Biosciences, Franklin Lakes, USA) and EDTA (K2F, BD #368861, BD Biosciences) tubes by a protocol approved by the departmental ethical board."

Once you have amended this statement in the Methods section of the manuscript, please add the same text to the “Ethics Statement” field of the submission form (via “Edit Submission”).

Reviewers' comments:

Reviewer's Responses to Questions

**Comments to the Author**

1. Is the manuscript technically sound, and do the data support the conclusions?

Reviewer #1: Yes

Reviewer #2: Yes

2. Has the statistical analysis been performed appropriately and rigorously? 

Reviewer #1: Yes

Reviewer #2: No

3. Have the authors made all data underlying the findings in their manuscript fully available?

Reviewer #1: Yes

Reviewer #2: Yes

4. Is the manuscript presented in an intelligible fashion and written in standard English?

Reviewer #1: Yes

Reviewer #2: Yes

5. Review Comments to the Author

Reviewer #1: Extracellular vesicles (EVs) have been extensively usedfor biomarker studies. It is expected that preanalytical processing methods of freshly draw blood may affect components of EV cargo. So far, however, few studies have evaluated such effect. To address this question, in this paper, Palviainen et al examined effect of anti-coagulation on isolated EV population and found significant increase on EV counts in serum when compared to plasma samples. Proteomics analysis showed significant differences between serum and plasma as well as among plasma samples with different anticoagulants. Results from the study enriches our knowledge on the effect of blood sample preparation methods on isolated EV characteristics. The manuscript is well-written, and the data are solid. It can be further improved by addressing the following questions:

1. The study mainly focused on difference. It will be beneficial to include scatter plots showing similarity (with r values) in commonly detected proteins.

2. Please provide a supplementary table showing proteomics data categorized by sample types (serum, different anticoagulants).

3. Keratin proteins are likely from contamination during sample preparation. A discussion on its origination should be added.

4. Discussion on limitation of the study should be added.

5. Figures are in bad quality in pdf file. High resolution in published manuscript is needed.

Reviewer #2: Palviainen and colleagues have conducted an interesting and useful work assessing how several anticoagulation reagents for plasma preparation and serum affect the concentration, cellular origin, and protein content of circulating extracellular vesicles (EV). The authors additionally and originally assessed, at least through bioinformatic methods, whether the detected protein cargo could include an EV protein signature resembling a reported protein corona from synthetic nanoparticles. The study is rigorous in its design, using state-of-the-art methods and it is clearly presented. However, there are several issues that need to be addressed to reach publication quality.

Major Issues:

1. The Statistical Analysis section (and statistical references throughout other items of the Materials and Methods section) needs major improvement. The authors need to clarify how they assessed the normality of the data before implementing ANOVA followed by Tukey tests, which assume normally distributed data. The statement “For multiple parameter analysis of non-normal distributed data Tukey multiple comparison test was used”, on line 194, is not accurate. What the authors need to do to handle comparisons of multiple groups of non-normal data is a transformation of the data to approximate normality before conducting an ANOVA test followed by the post-hoc Tukey test for multiple comparisons, or, alternatively, the implementation of a non-parametric test such as the Kruskal-Wallis test followed by appropriate post-hoc analysis such as the Dunn test. Additionally, information on the PCA approach is necessary.

2. Starting on page 18 of the result section “Protein cargo of the EV corona is modified by anticoagulation or its absence”, the authors described several comparisons with lists of proteins from published studies and claim (e.g., on line 322) that the overlap denote “marked similarities regarding the EV proteome”. For this claim to be justified, the statistical significance of the overlap between the different studies needs to be assessed. This can be done by calculating an empirical P value for the overlap among the protein lists based on simulation analyses. The simulation analysis would assess the probability of producing an equivalent random overlap between the published MP corona protein signatures (from the published studies) and a number of simulated protein lists (e.g., 1,000 or 10,000 simulated lists) that are generated by randomly selecting lists of proteins from Exocarta or EVpedia (with the simulated protein lists having the same length as the lists of experimentally detected EV proteins in this study). Only then could the overlap be claimed to be significant. The simulation analysis will reveal whether the detected overlap among the lists could have just happened by chance, therefore potentially not meaningful.\\

Minor Issues:

1. On line 90, the word “each” is misplaced.

2. Methods for EV isolation need to be better described. It is not clear what was the initial volume of pooled plasma or serum used for set I and II (the pellets of which were resuspended in 250 uL of PBS). It is not clear whether the elution from the SEC column was performed as a single elution step with 1 mL of PBS or whether there were multiple fractions collected.

3. Supplementary Figure 1 should be moved to the main text and significant digits used on Sup. Fig. 1A should be consistent.

4. Provide data table or representative NTA plots of size distributions for visual assessment of the size distribution differences instead of referring “data not shown” on line 215. This data could be included in the Supplementary Information document.

5. The statement “The generally upheld view of the EV field is that there are more platelet-derived “microparticles” or EVs, in serum than plasma,” on line 240 seems too subjective and needs a published reference. The sentence should be rephrased to more scientifically covey its meaning. The authors could instead write something similar as in the discussion section lines 348-349, e.g., “The fact that large EVs from serum have been shown to contain more platelet-derived microparticles than large EV from ACD plasma (7), prompted us to…”

6. Quality of all figures is unacceptably low. Replace figures with higher resolution versions.

7. Table 1 should go to Supplementary Information and columns ACD, Citrate, and EDTA clearly marked as P values.

8. The word “blotted” on line 275 should read “plotted”.

9. The statement: “At the moment, very few studies have addressed the factors (age, sex) which may affect EV concentrations in healthy populations”, on lines 362-363, requires citation(s) of some of the referenced studies.

6. PLOS authors have the option to publish the peer review history of their article (what does this mean?). If published, this will include your full peer review and any attached files.

Reviewer #1: No

Reviewer #2: Yes: Richard Pratley, MD

---

## [Author Response · Author response to Decision Letter 0]

14 Jun 2020

Response to reviewers

Reviewer #1: Extracellular vesicles (EVs) have been extensively used for biomarker studies. It is expected that preanalytical processing methods of freshly draw blood may affect components of EV cargo. So far, however, few studies have evaluated such effect. To address this question, in this paper, Palviainen et al examined effect of anti-coagulation on isolated EV population and found significant increase on EV counts in serum when compared to plasma samples. Proteomics analysis showed significant differences between serum and plasma as well as among plasma samples with different anticoagulants. Results from the study enriches our knowledge on the effect of blood sample preparation methods on isolated EV characteristics. The manuscript is well-written, and the data are solid. It can be further improved by addressing the following questions:

We would like to thank the reviewer for his or her positive remarks.

1. The study mainly focused on difference. It will be beneficial to include scatter plots showing similarity (with r values) in commonly detected proteins. 

We thank the reviewer for this valuable comment. We have performed Pearson correlation analysis (as described in Methods lines 199-200), and results are presented in Results (lines 284-286) and Supplementary files (S5 table).

2. Please provide a supplementary table showing proteomics data categorized by sample types (serum, different anticoagulants). 

We have now added the proteomics file as a Supplementary file (S4 table).

3. Keratin proteins are likely from contamination during sample preparation. A discussion on its origination should be added. 

We agree that the majority of the identified keratins likely originate from exogenous sources, and this information has been added to the Legend text of Table 1.

4. Discussion on limitation of the study should be added. 

We thank the reviewer for this valuable comment. We have now included the following limitations to the Discussion: the use of pooled samples, the use of differential ultracentrifugation, and the absence of analysis of the impact of the anticoagulation on the smallest EV. The comments have been integrated in the text and therefore the Discussion had to be rewritten and restructured to reorganize the text. The additions are highlighted.

5. Figures are in bad quality in pdf file. High resolution in published manuscript is needed. 

We wholly agree with both reviewers. The cause is the PDF-generating system of the journal, which produces low-resolution files. All figures accessible through the separate downloadable link were and still are in high resolution. We have now rechecked that all the figures meet the requirements of the journal.

 

Reviewer #2: Palviainen and colleagues have conducted an interesting and useful work assessing how several anticoagulation reagents for plasma preparation and serum affect the concentration, cellular origin, and protein content of circulating extracellular vesicles (EV). The authors additionally and originally assessed, at least through bioinformatic methods, whether the detected protein cargo could include an EV protein signature resembling a reported protein corona from synthetic nanoparticles. The study is rigorous in its design, using state-of-the-art methods and it is clearly presented. However, there are several issues that need to be addressed to reach publication quality.

Major Issues:

1. The Statistical Analysis section (and statistical references throughout other items of the Materials and Methods section) needs major improvement. The authors need to clarify how they assessed the normality of the data before implementing ANOVA followed by Tukey tests, which assume normally distributed data. The statement “For multiple parameter analysis of non-normal distributed data Tukey multiple comparison test was used”, on line 194, is not accurate. What the authors need to do to handle comparisons of multiple groups of non-normal data is a transformation of the data to approximate normality before conducting an ANOVA test followed by the post-hoc Tukey test for multiple comparisons, or, alternatively, the implementation of a non-parametric test such as the Kruskal-Wallis test followed by appropriate post-hoc analysis such as the Dunn test. Additionally, information on the PCA approach is necessary.

We thank the reviewer for the comment and valuable suggestions. Accordingly, we have now tested the data of all experiments for normality using Saphiro-Wilk test, and based on the results analyzed the data of flu-SEC experiment with Tukey test, and the flow cytometry and proteomics data using Kruskall-Wallis test. The statistics-paragraph in the methods has been rephrased based on these changes (lines 192-201). In addition, PCA is now explained in more detail (lines 277-284).

Based on the new statistical analysis of proteomic data done with Kruskall-Wallis test followed by Dunn test (post-hoc), the results were slightly changed. 35 proteins showed significant difference between different sample types and this has now stated in lines 306-314) and table 1 has been now corrected accordingly.

2. Starting on page 18 of the result section “Protein cargo of the EV corona is modified by anticoagulation or its absence”, the authors described several comparisons with lists of proteins from published studies and claim (e.g., on line 322) that the overlap denote “marked similarities regarding the EV proteome”. For this claim to be justified, the statistical significance of the overlap between the different studies needs to be assessed. This can be done by calculating an empirical P value for the overlap among the protein lists based on simulation analyses. The simulation analysis would assess the probability of producing an equivalent random overlap between the published MP corona protein signatures (from the published studies) and a number of simulated protein lists (e.g., 1,000 or 10,000 simulated lists) that are generated by randomly selecting lists of proteins from Exocarta or EVpedia (with the simulated protein lists having the same length as the lists of experimentally detected EV proteins in this study). Only then could the overlap be claimed to be significant. The simulation analysis will reveal whether the detected overlap among the lists could have just happened by chance, therefore potentially not meaningful.

We thank the reviewer for the insightful comment and advice concerning performing a simulation analysis. Firstly, we would like to point out that by the chosen wording (e.g. “markedly” in the Abstract) we tried to highlight the novelty, but not statistical significance of our finding as this was the first report on the similarities of EVs and NPs that we are aware of. However, we looked into this suggestion and consulted the experts available to us. Although we agree that a simulation would be very interesting to perform, we feel that in the scope of the current study it is too extensive and difficult for us to do without experience and absence of help on the matter. We also find that the databases on EVs are not well-curated regarding the experimental methods (e.g. EV isolation), so comparison against a randomly produced datasets of very variable studies are likely not be meaningful. Therefore, based on the reviewer’s comment, we moderated our text as follows: 

•the sentence in line 334-336 as following: “Based on the similarities of the EV proteome of our study and the two earlier studies using different isolation procedures, we hypothesize that there may be a common “EV protein corona”. 

•the sentence in lines 399-400 was adjusted and the word “remarkable” was removed. 

•in the Abstract “markedly” was removed, and the sentence now reads: “Comparison to other studies further revealed that the shared EV protein composition resembles the “protein corona” of synthetic nanoparticles incubated in plasma or serum.”

Finally, we were glad to discover a supporting publication from Simonsen and Münter, which was published while our work was being reviewed, which shares our concern about the similarities of NPs and EVs including the protein cargo they carry. “Pay attention to the biological nanoparticles when studying the protein corona on nanomedicines” (https://onlinelibrary.wiley.com/doi/abs/10.1002/anie.202004611). Although, the authors did not provide experimental data, we added this reference into the manuscript in the text (lines 406-407).

Minor Issues:

1. On line 90, the word “each” is misplaced.

The word has been removed.

2. Methods for EV isolation need to be better described. It is not clear what was the initial volume of pooled plasma or serum used for set I and II (the pellets of which were resuspended in 250 uL of PBS). It is not clear whether the elution from the SEC column was performed as a single elution step with 1 mL of PBS or whether there were multiple fractions collected.

The information on initial volumes of pooled samples for sets I and II is now added to the S1 table (workflow).

The elution from the SEC column was performed as a single elution step with 1 mL PBS. Column geometry and elution protocol was adapted from the application notes of PD MidiTrap G-25 columns (GE Healthcare), except that the stationary phase was replaced from Sephadex G-25 to Sepharose CL-2B. This protocol enables the separation of EVs from soluble proteins (e.g. albumin), but does not separate all lipoprotein particles. This has now been clarified in the manuscript (line 126). 

3. Supplementary Figure 1 should be moved to the main text and significant digits used on Sup. Fig. 1A should be consistent.

S1 figure has now been corrected, but in our opinion the S1 figure serves its purpose for providing visual data of background data of the experiment best in the supplementary files.

4. Provide data table or representative NTA plots of size distributions for visual assessment of the size distribution differences instead of referring “data not shown” on line 215. This data could be included in the Supplementary Information document.

A Figure depicting size distributions has now been added to S1 figure (C) and the sentence in lines 218-221 has been corrected accordingly.

5. The statement “The generally upheld view of the EV field is that there are more platelet-derived “microparticles” or EVs, in serum than plasma,” on line 240 seems too subjective and needs a published reference. The sentence should be rephrased to more scientifically covey its meaning. The authors could instead write something similar as in the discussion section lines 348-349, e.g., “The fact that large EVs from serum have been shown to contain more platelet-derived microparticles than large EV from ACD plasma (7), prompted us to…”

We thank the reviewer for this suggestion. The sentence has now been edited (lines 243-244).

6. Quality of all figures is unacceptably low. Replace figures with higher resolution versions.

Unfortunately, the PDF-file that the journal automatically creates contains a low-resolution version of the figures. All the figures were downloadable in high resolution format for the review from a separate toggle. We have rechecked and all figures meet the requirements of the journal. 

7. Table 1 should go to Supplementary Information and columns ACD, Citrate, and EDTA clearly marked as P values.

We are happy to remove the Table to Supplementary files if the reviewer insists that this will improve the paper. However, we would prefer to keep it in the main text, so that the information of the corona proteins would be easily accessible to the readers, which is not the case with separate supplementary information which is often accessed only a few people. We would like the readers to reflect on the meaning and impact of the selection of the corona proteins on EVs. Table 1 has been corrected after the new statistical analysis with Kruskall-Wallis test followed with Dunn´s test. As reviewer has suggested, the columns ACD, Citrate and Serum has now been marked with “p-value”. 

8. The word “blotted” on line 275 should read “plotted”.

Corrected

9. The statement: “At the moment, very few studies have addressed the factors (age, sex) which may affect EV concentrations in healthy populations”, on lines 362-363, requires citation(s) of some of the referenced studies.

We have added three references (lines 375-380): Eitan et al., Age-related changes in plasma extracellular vesicle characteristics and internalization by leukocytes. Scientific Reports 2017, Arraud et al., Extracellular vesicles from blood plasma: determination of their morphology, size, phenotype and concentration. Journal of thrombosis and haemostasis 2014, and Frühbeis et al., Physical exercise induces rapid release of small extracellular vesicles into the circulation. Journal of extracellular vesicles 2015.

---

## [Decision Letter · Decision Letter 1]

8 Jul 2020

Extracellular vesicles from human plasma and serum are carriers of extravesicular cargo ─ implications for biomarker discovery

PONE-D-20-08075R1

Dear Dr. Siljander,

We’re pleased to inform you that your manuscript has been judged scientifically suitable for publication and will be formally accepted for publication once it meets all outstanding technical requirements.

Kind regards,

John Matthew Koomen, PhD

Academic Editor

PLOS ONE

Additional Editor Comments (optional):

Reviewers' comments:

Reviewer's Responses to Questions

**Comments to the Author**

1. If the authors have adequately addressed your comments raised in a previous round of review and you feel that this manuscript is now acceptable for publication, you may indicate that here to bypass the “Comments to the Author” section, enter your conflict of interest statement in the “Confidential to Editor” section, and submit your "Accept" recommendation.

Reviewer #1: All comments have been addressed

Reviewer #2: All comments have been addressed

2. Is the manuscript technically sound, and do the data support the conclusions?

Reviewer #1: Yes

Reviewer #2: Yes

3. Has the statistical analysis been performed appropriately and rigorously? 

Reviewer #1: Yes

Reviewer #2: Yes

4. Have the authors made all data underlying the findings in their manuscript fully available?

Reviewer #1: Yes

Reviewer #2: Yes

5. Is the manuscript presented in an intelligible fashion and written in standard English?

Reviewer #1: Yes

Reviewer #2: Yes

6. Review Comments to the Author

Reviewer #1: The authors have satisfactorily addressed my comments. The data presented will fill some knowledge gap in EV research.

Reviewer #2: (No Response)

7. PLOS authors have the option to publish the peer review history of their article (what does this mean?). If published, this will include your full peer review and any attached files.

Reviewer #1: **Yes: **Liang Wang

Reviewer #2: **Yes: **Richard Pratley, MD

---

## [Editor Report · Acceptance letter]

24 Jul 2020

PONE-D-20-08075R1 

Extracellular vesicles from human plasma and serum are carriers of extravesicular cargo ─ implications for biomarker discovery 

Dear Dr. Siljander:

I'm pleased to inform you that your manuscript has been deemed suitable for publication in PLOS ONE. Congratulations! Your manuscript is now with our production department. 

Kind regards, 

on behalf of

Dr. John Matthew Koomen 

Academic Editor

PLOS ONE